# Estimating the Cost of Biofuel Use to Mitigate International Air Transport Emissions: A Case Study in Palau and Seychelles

**Yijun Hong [1,2], Huijuan Cui [1,]\*** **, Junhu Dai [1] and Quansheng Ge [1]**

[1]  Key Laboratory of Land Surface Patterns and Simulation, Institute of Geographic Sciences and Natural Resources Research, CAS, Beijing 100101, China
[2]  University of Chinese Academy of Sciences, Beijing 100049, China
\*  Correspondence: cuihj@igsnrr.ac.cn; Tel.: +86-10-6488-8895

**Abstract:** International air transport is one of the fast-growing sources of $CO_2$ emissions. However, it has always been omitted from the international emission mitigation pledges. The delayed mitigation process in this area may slow down the process of global $CO_2$ emission control. In this article, we evaluated the potential to realize the emission mitigation targets in air transport through biofuel and estimated the corresponding cost. The emission from international air transport of Palau and Seychelles was taken as the example. Then, the emission caused by each airline to these two islands was calculated by the distance-based method, with information of the travelers' arrival data, fuel consumption of different aircraft types, routes, and aircraft seat data. Future scenarios with and without commitment to $CO_2$ mitigation targets were predicted to evaluate the emission difference. Then, we estimated the amount of biofuel required to fill the emission gap, and the corresponding cost based on the future biofuel price prediction. The results show that distance is the determining factor of international air transport emission per capita. The component of origin can decrease the aggregated emission per capita to small island destinations by 0.5–2%. The accumulated emission gaps are 3.15 Mt and 9 Mt for Palau and Seychelles, which indicates that 7.64 and 19.34 Mb of biofuel are needed for emission mitigation, respectively. The corresponding costs are \$27–163 million and \$72–424 million per year.

**Keywords:** air transport; $CO_2$ mitigation; small islands; biofuel; cost

## 1. Introduction

International air transport emissions constitute 1.4% of the global greenhouse gas (GHG) emissions. Although the percentage is low, its rapid growth cannot be ignored, as air transport will become the dominant method for international transport by 2030 [1–3]. Olsthoorn X [4] revealed that $CO_2$ emission from international aviation may increase by a factor of 3–6 between 1995 and 2050. The International Air Transport Association (IATA) has proposed three emission mitigation targets after the 2008 Copenhagen conference. It was committed that the emission from international air-transportation would become carbon-neutral in 2020 and reduced to half of the 2005 net $CO_2$ emission in 2050 by increasing the ratio of air-transportation fuel efficiency by 1.5% per year [5]. Later, the International Civil Aviation Organization (ICAO) agreed on these targets and expected a more positive increase in ratio of fuel efficiency (2%) [1]. However, the current situation indicates that the emission mitigation lags behind the 2020 carbon-neutral goal if no more rigorous mitigation approach is taken [5], not to mention the long-term target. Besides, decreasing the international air transport emission is still a controversial area in the global mitigation roadmap [6]. The delay in the mitigation schedule may accelerate global

warming and lead to rising sea levels, health problems, economic risks or related environmental damages [7,8]. Therefore, it is urgent to seek other possible solutions for sustainable international air transport.

Biofuel, with the ability to reduce around 80% $CO_2$ emission compared to conventional jet fuel, is considered as the most promising solution for future sustainable international air transport, or even as the only real option to achieve the mitigation goals by 2050 [9]. The EU Renewable Energy Directive states that the mitigation potential of different types of biofuel varies from 40% to 104% [10,11]. El Takriti et al. reported that biofuel from lignocellulosic and waste feedstocks consistently provide substantial reduction in emissions [12]. However, the expensive price limits the use of biofuel, which costs more than 2–40 times than conventional jet fuel depending on the source of biofuel [6,13]. Deane analyzed the biofuel cost for EU, and indicated that the additional costs to achieve the 2050 mitigation target using bio jet fuels are between 0.42 €/L and 1.20 €/L [14]. El Takriti et al. indicated that biofuel will cost 2–6 times more than the conventional jet fuel depending on the feedstocks, and actual purchase costs will be several times higher than the estimated costs [12]. Although these studies have made specific analyses on prices of different biofuels, they did not combine the biofuel cost with international air transport emission mitigation needs. Therefore, the cost of achieving international air transport emission mitigation targets with biofuel needs further discussion.

The objective of this article is to investigate the possibility of mitigating emissions caused by international air transport by replacing jet fuel with biofuel. To estimate the amount of biofuel needed to fill the mitigation target, the distance-based method developed by Gössling and Scott [15] is used, which determines air transport emission based on the flight information, such as aircraft type, flight distance, and number of passengers. However, these data are not always accessible for analyzing the international air transport emission. Therefore, this study carried out two case studies in Palau and Seychelles, where are the two tourism-reliant island countries, for the simple reason that 88–91% of their international air transport arrivals were occupied by the international tourists, which have completely recorded in their year book. In this study, three future scenarios, namely the baseline scenario, technological mitigation scenario, and target scenario, were assumed to estimate the amount and corresponding cost of biofuel needed to fill the mitigation gap.

## 2. Materials and Methods

### 2.1. Data Description

In order to estimate $CO_2$ emissions from air transport, the fuel consumption of each international flight from its origin to Palau or Seychelles needed to be determined, which can be calculated by multiplying the international arrival number and fuel consumption of each passenger. Therefore, the number of passengers on each flight route, the origins of the passengers, fly distance of each route and air craft type, are needed.

The international arrival data were obtained from yearbooks of Palau (http://palaugov.pw/rop-statistical-yearbooks/) and Seychelles (http://www.nbs.gov.sc/statistics/tourism), from which we found that the most passengers by international air transport to Palau and Seychelles are international tourists with the percentage over 88–91%, even over 95% and 93% in 2015, respectively. Therefore, the international tourist numbers can represent the total international passengers in both island countries. The original countries of international passengers to two islands are the US mainland, Canada, Europe, Australia, Korea, Japan, China, Philippines, Singapore, etc. For the convenience of calculation, the passengers were categorized into corresponding continents. For example, the Palau passengers were concluded as originating from Asia, North America, Europe, Oceania, and others; while the Seychelles passengers came from Asia, America, Africa, Europe, and Oceania.

Fuel consumption is based on the data of flying distance, aircraft types, theoretical fuel consumption by distance of different aircraft types, theoretical number of passenger seats in all economy class of different aircraft types, passenger to freight factor, and passenger load factor of each route. The flying

distance is a corrected Great Circle Distance (GCD) [16] between two paired airports. Thus, the specific latitude and longitude of the airports are needed, which can be obtained from the Open Flights dataset (https://openflights.org/). Moreover, this dataset also includes aircraft types in every route. We searched for information of flights arriving at Babelthuap Airport in Palau (ROR) and Seychelles International Airport (SEZ) in Seychelles, which are the only international airports of these islands (Tables A1 and A2).

The theoretical fuel consumption data by distance for different aircrafts related to the two islands' airlines were collected from ICAO and plotted in Figure 1. The theoretical number of passenger seats in all economy class was obtained from the Airplane Characteristics for Airport Planning, published by Boeing (http://www.boeing.com/commercial/airports/plan_manuals.page) and Airbus (http://www.airbus.com/support-services/airport-operations/aircraft-characteristics/), who have manufactured most of the commercial aircrafts serving the world currently. The passenger to cargo factor and passenger load factor, which reflect the real load in each route, can be obtained from the Load Factors by Route Group, published by ICAO, which includes the factors between certain regions around the world, based on the data collected by ICAO.

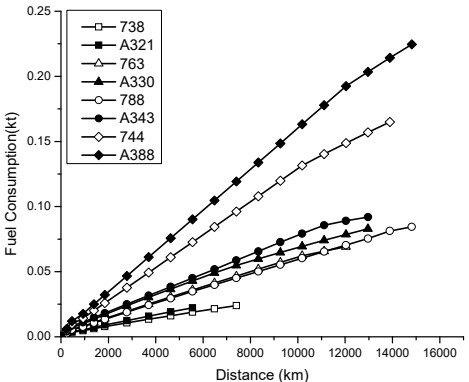

**Figure 1.** Theoretical fuel consumption of different aircrafts by flight distance.

The cost to fulfill the mitigation targets in the future is based on the data of conventional fuel price. Therefore, the fuel consumption and cost for the airline industry from 2005 to 2016 were collected from the IATA [17], which are listed in Table 1.

**Table 1.** Airline industry conventional fuel consumption and cost.

| Year | Conventional Fuel, $ Billion | Fuel Consumption, Billion Gallons |
|---|---|---|
| 2005 | 91 | 68 |
| 2006 | 127 | 69 |
| 2007 | 146 | 71 |
| 2008 | 203 | 70 |
| 2009 | 134 | 66 |
| 2010 | 151 | 70 |
| 2011 | 191 | 72 |
| 2012 | 228 | 73 |
| 2013 | 231 | 74 |
| 2014 | 224 | 77 |
| 2015 | 175 | 81 |
| 2016 | 133 | 85 |

## 2.2. Methodology

### 2.2.1. $CO_2$ Emission of Islands' International Air Transport

The international air transportation emissions of Palau and Seychelles were calculated by the distance-based method, which was developed by Gössling et al. [15]. The method involves adding up the per capita emission from each origin. Gössling's method uses geo-centroid determination of the average distances between international tourism markets, which were updated with more accurate locations of the airports with the data mentioned in 2.1. The $CO_2$ emission amount of two islands can be calculated by the following equation:

$$C_t = \omega \sum_{i=1}^{n} N \sum \frac{F\alpha}{Y\beta} \tag{1}$$

where:

$\omega$ is the transfer ratio of air transport fuel to $CO_2$ per ton, which is 3.16 in this study.
n is the number of the continents the passengers come from.
N is the passenger number coming from one continent.
F is the fuel consumption of the aircrafts serving this route.
$\alpha$ is the luggage load factor of the route. This ratio can influence the real fuel consumption in each route.
Y is the one-class seat of the aircraft flying this route.
$\beta$ is the passage load factor of the route, which represents the average passage load percentage of each route.
$C_t$ is the air transport $CO_2$ emission caused by the passenger visit from all over the world.

In Equation (1), $F\alpha/Y\beta$ is the fuel consumption per person of one route, which multiplied by N yields the total fuel consumption of this route. With all the routes summed up, the total fuel consumption for travel to one direction can be calculated. The emission generated by this international air travel can be obtained by multiplying the emission factor $\omega$. It should be noted that due to limited airline arrangements to the tourism destination [18], some passengers must take multiple trips to get to Palau and Seychelles. For example, passengers from Europe and Oceania have to make one stop in one of the Asian airports (TPE, NRT, MFM, ICN, MNL). Therefore, the distance for passengers from Europe is calculated as the average distance from LHR (London Heathrow Airport) airport to the available Asian airports, plus the distance from Asia to Palau. On the other hand, passengers from North America need to transfer twice, which means the distance from America to Palau is accumulated from several sections. The distances from JFK or SFO airports to HNL (Hawaii), then the distance from HNL (Hawaii) to GUM (Guam Island), and finally the distance from GUM to ROR (Palau) have to be summed as a whole. For Seychelles, the distance is calculated in the same manner as for Palau.

### 2.2.2. Prediction of Future International Air Transport Emission

To predict future emissions, three scenarios were assumed: The baseline scenario, technological mitigation scenario, and target scenario.

The baseline scenario is the future emission situation if no action is taken to mitigate the international air transport emission. The baseline scenarios of these two islands are based on the growing ratio of passengers multiplied by emission per passenger. The growing ratio of passengers around 3% annually as predicted by UNWTO (United Nations World Tourism Organization) [2], while the emission per passenger is taken consistently as the most recent value of emission per passenger.

The technological mitigation scenario indicates the future with jet fuel efficiency improved by the methods other than biofuel, including technology improvement and optimization of management and operation. For computation of these two islands, the fuel efficiency is set to increase by 2% per year, which is the most optimistic ratio [19].

The target scenario involves making the international air transport emission per kilometer decrease by 20% below 2013 level in 2020 [20] and decrease to half of the 2005 emission in 2050. The mitigation potential to be filled by biofuel is the gap between the target scenario and the technology mitigation scenario.

### 2.2.3. Estimation of Future Biofuel Costs

Assuming that the production of biofuel will not be limited, the financial need can be estimated by multiplying the needed biofuel amount by the future predicted biofuel price (p). The future biofuel price can be calculated as:

$$p = (2 \sim 7)\frac{P_t}{P_c}P_c \qquad (2)$$

where

P is the biofuel price.
$P_t$ represents conventional jet fuel price.
$P_c$ represents crude oil price. The future conventional jet fuel is obtained from the World Bank's crude oil prediction [21].
The term (2~7) means that biofuel price is 2–7 times higher than conventional jet fuel.

## 3. Results

### 3.1. Historical Record of International Air Transport Emission of Palau and Seychelles

The international air transport passenger numbers have increased over time in both island countries, indicating the booming of tourism sector. As shown in Figure 2, number of passengers to Palau increased from 64,000 in 1998 to 162,000 in 2015, while that to Seychelles increased from 130,046 in 2000 to 303,177 in 2016. The average GCD values from the paired airports in different continents are plotted in Figure 2b, where the direction of arrows shows the passengers' origin, and the colors of the arrows correspond to the different continents. In general, most of the passengers came from the nearest continents, as passengers from Asia and Europe formed the largest percentage of passengers to Palau (92%) and Seychelles (76%), respectively. The percentage of passengers coming from different continents remained almost the same. However, it was noted that the number of passengers from Asia to both Palau and Seychelles increased significantly.

Figure 3a,c shows the emission per capita of international air transport passengers to Palau and Seychelles from different continents. The colors of the arrows are consistent with the continent legend, and width of the arrow represents the emission amount. For Palau, the emission per capita of European passengers was the highest (1297 kg) while the lowest $CO_2$ emission (370 kg) was from each Asian passenger. For Seychelles, the emission per capita of American passengers ranked the first, and every passenger caused 2178 kg of $CO_2$ emission in their journey, while passengers from Africa caused the least emission per capita, of only 478 kg.

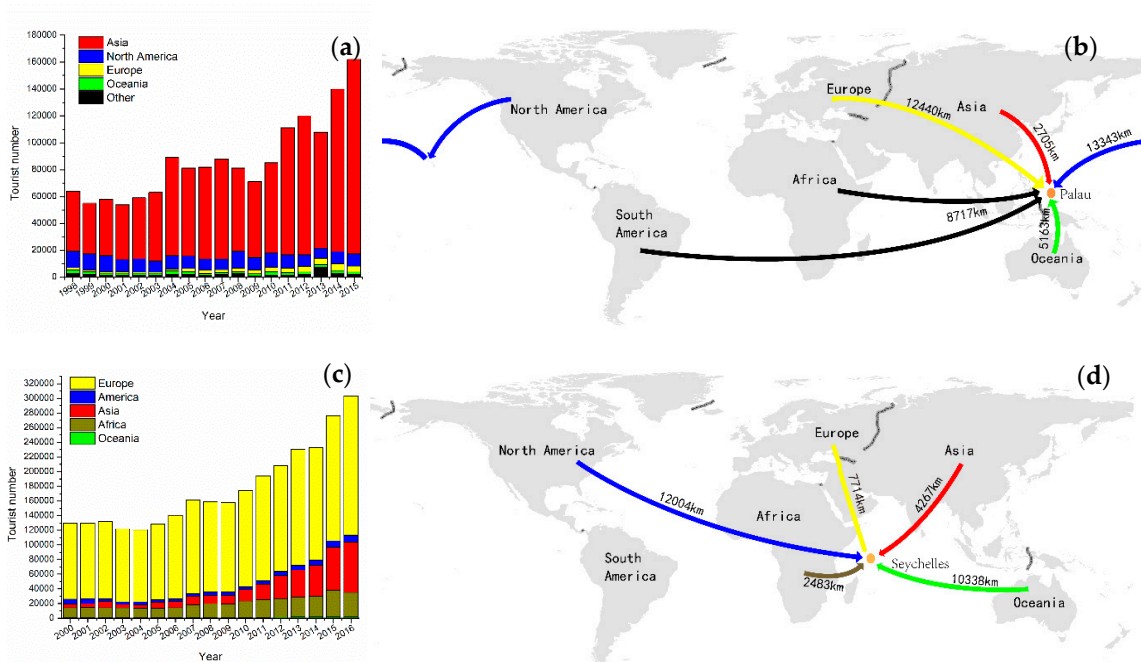

**Figure 2.** Historical passenger numbers (**a**,**c**) and the GCD (Great Circle Distance) (**b**,**d**) of Palau and Seychelles.

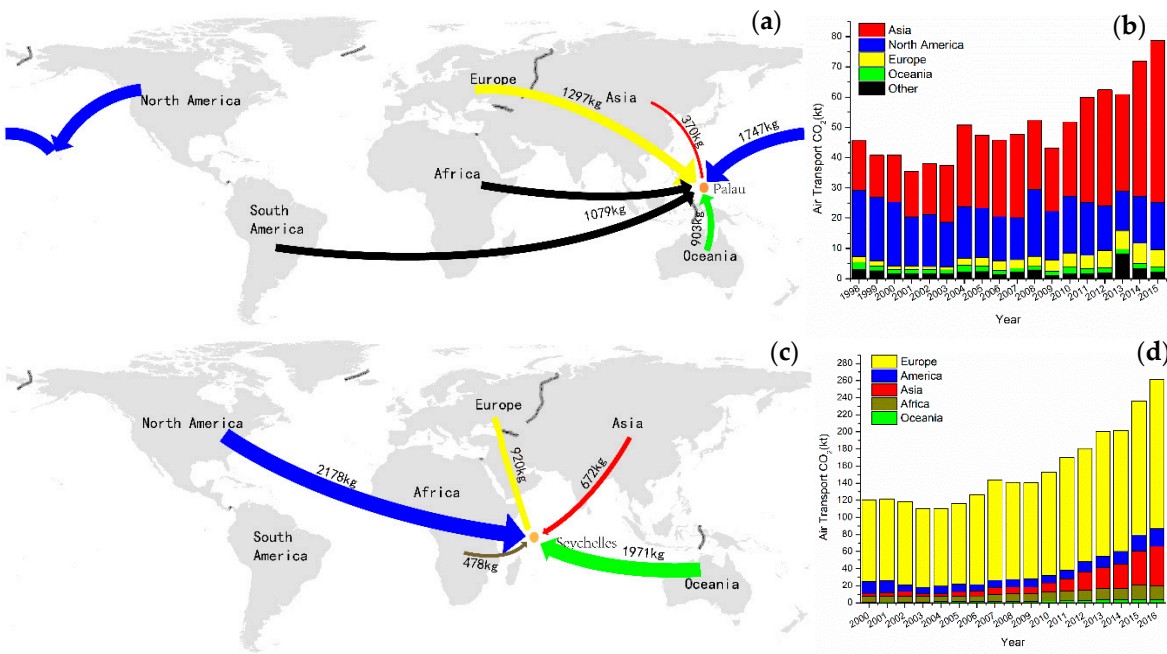

**Figure 3.** International air transport emission (**a**,**c**) and emission per capita by continents (**b**,**d**) of Palau and Seychelles.

Combining the information of number of passengers, distance and emission per capita, Figure 3 shows the estimated round-trip air transport emission of the two islands based on Equation (1). As shown in Figure 3b, the international air transport emission of Palau increased from 45.6 kt in 1998 to 78.7 kt in 2015 and the ratio of different origins changed. Although the passengers coming from North America were not the highest number, the emission caused by North America was the largest in 1998 due to the long distance of flying. As the number of Asian passengers increased, Asian air

transportation emission increased from 16.5 kt to 53.5 kt, which account for 67.99% of total air emission in 2015. As shown in Figure 3d, the international air transport emission of Seychelles increased from 120.6 kt in 2000 to 261.14 kt in 2016. European passengers were the major part of the international passengers, and the European international air transport emission and percentage ranked first during this period. Again, as the number of passengers coming from Asia increased, emission caused by Asian passengers increased the most. Specifically, their emission amount increased from 3.61 kt to 46.43 kt, and its percentage increased from 3% to 18%.

### 3.2. Prediction of International Air Transport Emission of Palau and Seychelles

Figure 4 shows the future international air transport scenarios of Palau and Seychelles. In Figure 4a, the black line represents the baseline scenario of Palau. Under this scenario, the aggregated international air transport emission per capita is consistent with the value in 2015. As the number of passengers increase, the emission will increase to 0.26 Mt in 2050. The grey area shows the mitigation contribution by technology and management, which is the most positive mitigation scenario assuming a 2% increase in mitigation efficiency per year. Under this scenario, the emission has to decrease to 0.19 Mt in 2050. The green area shows the gap to be filled through biofuel to achieve the mitigation targets. There is 0.05 Mt to fill in order to achieve the turning point in 2020. In 2050, the gap is extended to 0.18 Mt, and the accumulated mitigation gap from 2016 to 2050 is 3.15 Mt. Figure 4b shows the scenarios of Seychelles. Under baseline scenario, the international air transport emission will increase to 0.71 Mt in 2050. Under the most positive technology and management mitigation scenario, the emission will decrease to 0.53 Mt in 2050, which leaves a mitigation gap of 0.47 Mt to be filled by biofuel. Moreover, the accumulated mitigation gap from 2017 to 2050 is 3.15 Mt.

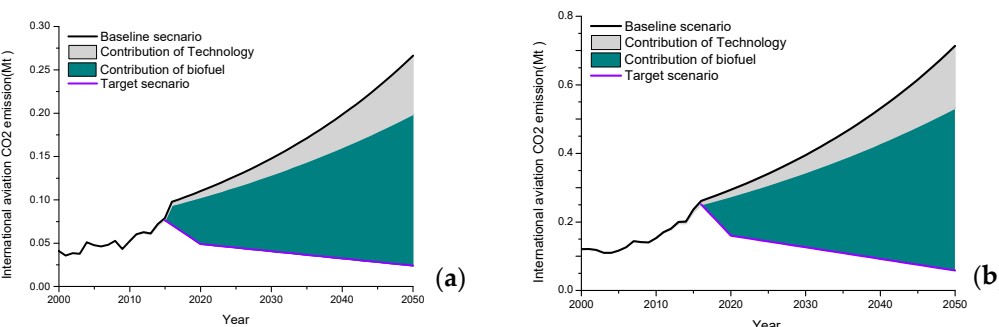

**Figure 4.** Predicted future international air transport emission under different scenarios of (**a**) Palau and (**b**) Seychelles.

### 3.3. Cost of Achieving Mitigation Targets of International Air Transport Emission through Biofuel

Based on the fuel consumption and fuel cost data in IATA report, the fuel price in airline industry is about 97 $/b (average data of 2005–2016, with crude oil and jet kerosene per barrel). The ratio between air transport industry oil and crude oil is around 1.2. Based on the crude oil forecast reported by the World Bank [22], the price of air transport fuel was estimated, as shown in Table 2. The unit $/b refers to the price per barrel. The conventional air transport fuel cost is expected to increase to 78.15 $/b in 2020, 88.21 $/b in 2025, and 99.39 $/b in 2030. Biofuel price is taken as 2–7 times the price of conventional air transport fuel. Therefore, the price of air transport biofuel would be 156–547 $/b in 2020, 176–617 $/b in 2025 and 199–696 $/b in 2030.

**Table 2.** Forecast price of air transport biofuel.

| Year | World Bank Crude Oil, $/b | Conventional Fuel Oil, $/b | Biofuel, $/b |
|------|---------------------------|----------------------------|--------------|
| 2017 | 55 | 68.33 | 137–478 |
| 2018 | 60 | 74.54 | 149–522 |
| 2019 | 61.5 | 76.41 | 153–535 |
| 2020 | 62.9 | 78.15 | 156–547 |
| 2021 | 64.5 | 80.14 | 160–561 |
| 2022 | 66 | 82 | 164–574 |
| 2023 | 67.6 | 83.99 | 168–588 |
| 2024 | 69.3. | 86.1 | 172–603 |
| 2025 | 71 | 88.21 | 176–617 |
| 2030 | 80 | 99.39 | 199–696 |
| 2050 | 137 | 170 | 340–1191 |

As shown in Table 3, the financial investment in biofuel needed to accomplish the emission mitigation targets of the two islands is expected to be about 8–48 m$ in 2020 and 71–425 m$ in 2050 for Palau, meaning 0.1 million barrels (Mb) of biofuel would be needed in 2020 and 0.42 Mb of biofuel would be needed in 2050. Moreover, the extra costs for Seychelles are 17–102 m$ in 2020 and 192–1150 m$ in 2030. Palau's total emission mitigation potential from 2016 to 2050 is 3.15 Mt, which would require 953–5721 m$, while Seychelles' emission mitigation potential is 9 Mt, which would require 2460–14,762 m$ investment.

**Table 3.** Emission mitigation potential and costs for international air transport of Palau and Seychelles.

| Year | Palau | | | Seychelles | | |
|------|-------------------------------|-------------------------|-------------------|-------------------------------|-------------------------|-------------------|
| | Mitigation Potential, Mt | Biofuel Demand, Mb | Extra Cost, m$ | Mitigation Potential, Mt | Biofuel Demand, Mb | Extra Cost, m$ |
| 2020 | 0.05 | 0.1 | 8–48 | 0.12 | 0.22 | 17–102 |
| 2025 | 0.07 | 0.14 | 12–73 | 0.17 | 0.32 | 29–171 |
| 2030 | 0.09 | 0.18 | 18–107 | 0.22 | 0.44 | 44–265 |
| 2050 | 0.18 | 0.42 | 71–425 | 0.47 | 1.13 | 192–1150 |
| Sum | 3.15 | 7.64 | 953–5721 | 9 | 19.34 | 2460–14,762 |
| Average | 0.09 | 0.22 | 27–163 | 0.26 | 0.57 | 72–434 |

*3.4. Biofuel Needs and Cost of Global International Air Transport Emission Mitigation*

Based on the results of fuel price prediction, the biofuel needs and corresponding cost of global international air transport emission mitigation were further calculated. The calculate method is the same as that for these two islands.

The global energy consumption data of international air transport was obtained from IEA. This dataset is the collection of energy consumption data of most countries in the world from 1971 to 2014 [23]. The data indicates that the energy type of international air transport has always been oil products, with no percentage of biofuels or other energy types. The global emission from international air transport was computed by multiplying the fuel consumption and emission factor (3.16). Figure 5 shows the global international air transport emission in 1971 to 2014, which increased from 178 Mt to 532 Mt. The red dotted line in Figure 5 is the fitted exponential function with $r^2 = 0.98$.

To predict the future baseline of global international air transport emission, the time series regression of recorded emission data was first fitted and then it was extended to 2050. The target scenario stands for the goals of both carbon-neutral in 2020 and half of the 2005 emission in 2050. For the global target scenario, the implementation of biofuel starts from 2020, leading to the 2050 target, as presented in Figure 5. The emission can be extended to 2050 by fitting past series, and is estimated to be 625 Mt in 2020 and 1384 Mt in 2050. In the technological mitigation scenario with the fuel efficiency increasing by 2% per year, the emission will be 585 Mt in 2020 and 1034 Mt in 2050. The target scenario is set to be 222 Mt in 2050, and the green area shows the mitigation gap, which is to be filled by biofuel.

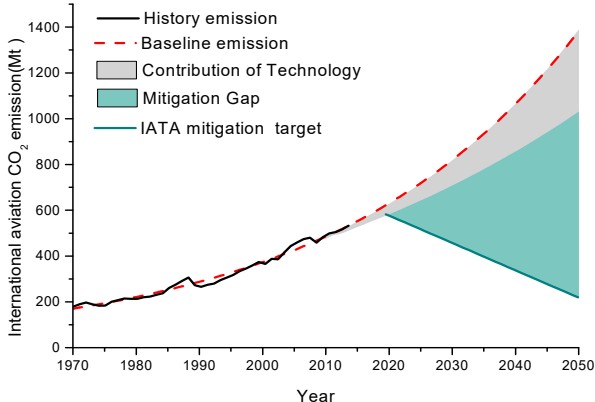

**Figure 5.** Predicted future scenarios of global international air transport emission.

The predicted mitigation potentials from the previous sections are 121 Mt in 2025, 247 Mt in 2030 and 811 Mt in 2050, which will require the financial investment of 21–125 billion dollars (b$) in 2025, 50–299 b$ in 2030, and 328–1967 b$ in 2050, as shown in Table 4. The mitigation potential to be filled from 2020 to 2050 is estimated to be 12,045 Mt, with 402 Mt required annually. This requires about 885 billion barrels (bb) of biofuel. Therefore, closing the gap with biofuel will require an investment of 3584–21,506 b$.

**Table 4.** Mitigation potential and extra costs of global international air transport using biofuel.

| Year | Mitigation Potential, Mt | Biofuel Demand, bb | Extra Cost, b$ |
|------|--------------------------|--------------------|----------------|
| 2025 | 121 | 237 | 21–125 |
| 2030 | 247 | 501 | 50–299 |
| 2050 | 811 | 1925 | 328–1967 |
| Sum | 12,045 | 26,545 | 3584–21,506 |
| Annual | 402 | 885 | 119–717 |

## 4. Discussion

In fact, there are four main mitigation pathways for air transport emission such as mitigation policy, technology improvement, operation and management, and alternate jet-fuel, within which, the contribution of operation and management is very limited. Besides, considering the commercial cost of the aircrafts, it is not a realistic solution to upgrade the aircraft very quickly unless the improved technology in airplane design or engine regeneration. As a result, other mitigation methods besides biofuel will not create significant changes in the coming century [24]. The European Aviation Environmental Report 2019 also states that sustainable aviation fuels have the potential to make an important contribution to mitigating the current and expected future environmental impacts of aviation [25]. For this reason, biofuel was considered as the major factor responsible for future emission mitigation. However, till now the potential of biofuel and the detailed approach, are yet to be discussed thoroughly [26]. This study also suffers from a lot of uncertainties, arising from the prediction of emission and corresponding cost [27].

As shown in Section 3.1, the main determining factor for the aggregated emission per capita is the distance—the longer the distance passengers are from the two islands, the greater the emission they will bring. However, it was found that the component of international air transport origin is also important to the aggregated emission per capita. With the increase in Asian passengers in recent years, Asia brings the second largest $CO_2$ emission, even though its emission per capita is the least among the five regions of origin. In Seychelles, although the emission per capita from America ranks the first, the total emission from America ranked the third in 2016 because of the low passenger number. Therefore, as shown in Figure 6, in Palau, the boost of Asian passengers led to the decrease in aggregated emission

per capita (2% per year). In Seychelles, on the other hand, the decrease in aggregated emission per capita was slower (0.5% per year) than that of Palau because of its consistent passenger origin component. This phenomenon is in accordance with the conclusion made by Gossling [26], that the emission of island tourism can be influenced by the management of market directions. However, in this study, the aggregated emission per capita was taken as a constant value when the future emission of these two islands was estimated, which may bring some uncertainties in the future emission prediction of the two islands.

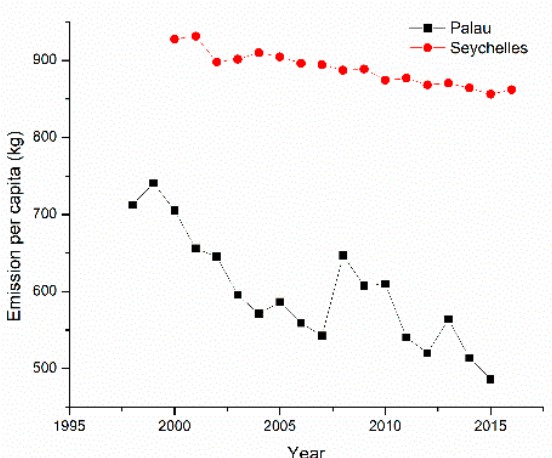

**Figure 6.** Aggregated international air transport emission per capita of the two islands.

When computing the biofuel demand to achieve the mitigation targets set by the ICAO, it was assumed that the biofuel supplementation is sufficient, which may not be true in actual situation. In fact, there are only four biofuel factories producing bio-jet fuel currently. Compared to bio-jet production, bio-diesel production is the major component of the factory's biofuel production, which makes the supplementation of biofuel for air transport more pessimistic. Moreover, there is still debate about the mitigation potential of increased biofuel in transportation. According to IREAN's report, the emission-reduction potential of different feedstocks may differ significantly, with values ranging from 50% to 95% of the claimed potential reduction when compared with fossil jet fuel [9]. Besides, the uncertainty of biofuel price is not considered in this work. Here, it was assumed that the biofuel price is 2–7 times of conventional fuel, but there are other ratio assumptions about the biofuel price. For instance, the European Aviation Environmental Report 2019 assumed that the biofuel price is 1.6~1.7 times that of conventional fuel. Furthermore, Pavlenko N stated that the price of alternative jet fuels is two to eight times the price of petroleum jet fuel. in the European Union [28]. Therefore, the cost estimated in this study contains a lot uncertainty, which can be reduced by further narrowing the range of the bio fuel price.

Furthermore, in the prediction of future baseline emission scenarios and mitigation scenarios, the trajectories are considered as the extrapolation of fitted trends. However, the future emission of international air transport to small islands can be easily influenced by multiple factors, such as the international economic situation and natural disasters.

## 5. Conclusions

In this paper, the characteristics of international air transport emission of Palau and Seychelles were analyzed, and the emission mitigation potential using biofuel was explored for these two islands and globally. Moreover, the corresponding cost of replacing conventional jet fuel with biofuel was estimated. The following conclusions can be drawn from our study:

1) Distance is the determining factor of the emission per capita caused by international air transport, while the component of origin decreases the aggregated emission per capita to small island destinations by 0.5%–2%.

2) Biofuel provides a possible pathway to realize the emission mitigation targets of international air transport. In order to mitigate emissions by 3.15 and 9 Mt for Palau and Seychelles, 7.64 and 19.34 Mb of biofuel are needed, respectively. The corresponding costs are 27–163 m$ and 72–424 m$ per year, respectively.

3) The global emissions mitigation target of international air transport in 2050 can be achieved using 885 bb of biofuel. The implementation of biofuel in international air transport can reduce 402 Mt of $CO_2$ emission per year, which will require the financial investment of 119–717 b$ per year.

**Author Contributions:** Y.H. and H.C. conceived and designed the structure of the paper, Y.H. analyzed the data, and all authors wrote the manuscript together.

**Acknowledgments:** The authors express their sincere thanks to the financial support from the national key research and development program of China (2017YFA0605303).

**Conflicts of Interest:** The authors declare no conflict of interest.

**Appendix A**

The three letters abbreviation in tables is the IATA code, each representing an airport. It should be noted that the location of each airport is given by its longitude and latitude. For latitude, negative is South, while positive is North, and for longitude, negative is West, while positive is East.

**Table A1.** Information of airports and aircraft types in Palau international route.

| Continent | Destination | Longitude and Latitude | Origin | Longitude and Latitude | Aircraft Type |
|---|---|---|---|---|---|
| Asia | ROR | 7.37, 134.54 | TPE | 25.08, 121.23 | 738 |
| | | | NRT | 35.76, 140.39 | 752, 763 |
| | | | MFM | 22.15, 113.59 | 739 |
| | | | ICN | 37.47, 126.45 | A321, 73H |
| | | | MNL | 14.50, 121.01 | 738 |
| North America | ROR | 7.37, 134.54 | GUM | 13.48, 144.80, | 738, 73G |
| | GUM | 13.48, 144.79 | HNL | 21.32, −157.92 | 777 |
| | HNL | 21.32, −157.92 | JFK | 40.64, −73.78 | A332, 76W |
| | | | SFO | 37.62, −122.38 | A320, A332, 738, 739, 753, 777 |
| Oceania | MNL | 14.51, 121.02 | MEL | −37.67, 144.84 | A330, A332 |
| | | | DRW | −12.41, 130.88 | A320, A321 |
| | ROR | 7.37, 134.54 | YAP | 9.50, 138.08 | 738, 73G |

**Table A1.** *Cont.*

| Continent | Destination | Longitude and Latitude | Origin | Longitude and Latitude | Aircraft Type |
|---|---|---|---|---|---|
| Europe | NRT | 35.76, 140.39 | LHR | 51.47, −0.46 | 789 |
| | MNL | 14.51, 121.02 | | | A343 |
| | ICN | 37.47, 126.45 | | | A388, 744, 74H, 77L, 77W, 788 |

**Table A2.** Information of airports and aircraft types in Seychelles international route.

| Continent | Destination | Longitude and Latitude | Origin | Longitude and Latitude | Aircraft Type |
|---|---|---|---|---|---|
| Asia | SEZ | −4.67, 55.52 | DXB | 25.25, 55.36 | 77W, 77L |
| | | | CMB | 7.18, 79.88 | A32A, A320 |
| | | | AUH | 24.43, 54.65 | A32A, A332, A320 |
| | | | IST | 40.98, 28.81 | A333, A332 |
| | | | BOM | 19.09, 72.87 | A332, A320 |
| | | | PEK | 40.08, 116.58 | A332 |
| | | | DOH | 25.26, 51.61 | A320, 319 |
| Europe | SEZ | −4.67, 55.52 | FRA | 50.03, 8.57 | 767 |
| | | | CDG | 49.01, 2.55 | A332 |
| | | | DUS | 51.29, 6.77 | A332 |
| Africa | SEZ | −4.67, 55.52 | NBO | −1.32, 36.93 | E90, 738 |
| | | | ADD | 8.98, 38.80 | 73w, 738, 763 |
| | | | RUN | −20.89, 55.51 | 738 |
| | | | MRU | −20.43, 57.68 | A332, A320 |
| | | | JNB | −26.14 | A332, A320 |
| | | | TNR | −18.80, 47.48 | A320 |
| | | | DUR | −29.61, 31.12 | A320 |
| America | DXB | 25.25, 55.36 | JFK | 40.64, −73.78 | A388 |
| | | | SFO | 37.62, −122.38 | A388 |
| Oceania | DXB | 25.25, 55.36 | MEL | −37.67, 144.84 | A388, 77w |
| | | | PER | −31.94, 115.97 | A388, 77W, 77L |

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
