# Peer review of "Estimating the Cost of Biofuel Use to Mitigate International Air Transport Emissions: A Case Study in Palau and Seychelles"

_sustainability, doi:10.3390/su11133545_

Round 1

Reviewer 1 Report

Interesting article, although it does not contain new views or analyze phenomena that have not been explored so far. Classical methods were used, but they were used properly and correctly presented.

Attention should be paid to the following:

1) The purpose has not been properly described in the introduction. In principle, it is not known exactly what the purpose of the article was and what was the purpose of the research?

2) Are the secondary data for the study separated into air transport and tourist air transport? (point 2.1)

3) The method described in point 2.2.1. refers to the aviation fuel consumption per capita. The calculations for fuel consumption in general have been applied. No fuel consumption was determined only for tourist flights.

4) The article compares the consumption of the total amount of fuel in air traffic on given routes in relation to the fuel consumption in tourist traffic. Air traffic is not only tourism in these directions. The same applies to carbon dioxide emissions. And biofuel consumption. The comparison made has no reason to exist.

5) The title is inadequate if the article analyzes the air tourism movement.

Author Response

Response to Reviewer 1 Comments

Point 1: The purpose has not been properly described in the introduction. In principle, it is not known exactly what the purpose of the article was and what was the purpose of the research?

Response 1: Thank you. The purpose of this article is to investigate the possibility of mitigating emissions caused by international air transport by replacing jet fuel with biofuel, which we might fail to clearly state in the first version. We have totally rewritten the introduction and stated the object at the end, and hope it is clearly explained this time.  

The last paragraph has revised as “The objective of this article is to investigate the possibility of mitigating emissions caused by international air transport by replacing jet fuel with biofuel. To estimate the amount of biofuel needed to fill the mitigation target, the distance-based method developed by Gössling and Scott [14] is used, which determines air transport emission based on the flight information, such as aircraft type, flight distance, and number of passengers. However, these data are not always accessible for analyzing the international air transport emission. Therefore, this study carried out two case studies in Palau and Seychelles, where are the two tourism-reliant island countries, for the simple reason that 88-91% of their international air transport arrivals were occupied by the international tourists, which have completely recorded. In this study, three future scenarios, namely the baseline scenario, technological mitigation scenario, and target scenario, were assumed to estimate the amount and corresponding cost of biofuel needed to fill the mitigation gap.”

Point 2: Are the secondary data for the study separated into air transport and tourist air transport? (point 2.1)

Response 2: Thank you for your suggestion. We have checked the purpose of international passengers to Palau and Seychelles and found that the most passengers by international air transport to Palau and Seychelles are found out to be international tourists with the percentage over 88-91%. Therefore, we did not separate data into air transport and tourist air transport, and roughly considered international air transport tourist number as the total international air transport passenger number for these two islands.

We added “The international arrival data were obtained from yearbooks of Palau (http://palaugov.pw/rop-statistical-yearbooks/) and Seychelles (http://www.nbs.gov.sc/statistics/tourism), from which we found that the most passengers by international air transport to Palau and Seychelles are found out to be international tourists with the percentage over 88-91%, even over 95% and 93% in 2015. Therefore, the international tourist number can represent the total international passengers in both island countries.” in Line 80 to make it clearer in the article.

Point 3: The method described in point 2.2.1. refers to the aviation fuel consumption per capita. The calculations for fuel consumption in general have been applied. No fuel consumption was determined only for tourist flights.

Response 3: Thank you. As we stated in the above responses, we did not mean to estimate the fuel consumption specifically to tourist flights but considered internarial tourist roughly as the international arrivals. Therefore, the general consumption calculation method should be appropriate to estimate the fuel consumption per capita.

Point 4: The article compares the consumption of the total amount of fuel in air traffic on given routes in relation to the fuel consumption in tourist traffic. Air traffic is not only tourism in these directions. The same applies to carbon dioxide emissions. And biofuel consumption. The comparison made has no reason to exist.

Response 4: Thank you. As we explained in the above responses, we considered international tourist roughly as the international arrivals. Therefore, tourism is taken as the only purpose of air traffic, so is the carbon dioxide emissions, and biofuel consumption. In conclusion, there is no comparison.

Point 5: The title is inadequate if the article analyzes the air tourism movement.

Response 5: Thank you. But we respectfully disagree with your point. The main purpose of this paper is to investigate the possibility of mitigating emissions caused by international air transport by replacing jet fuel with biofuel. The reason we analysed the tourism movement is that the most passengers by air transport to Palau and Seychelles are tourists. Therefore, we analysed their population and origins to estimate their CO2 emission. We are sorry for not making the point clearly in the first version, and have revised the manuscript thoroughly and hope this time the manuscript fit the scope of the title.

Reviewer 2 Report

The paper addresses an important issue that is relevant at this juncture. However, the main conclusion from the study, namely, "Distance is the determining factor of the emission per capita caused by international air transport, while the component of tourist origin can influence the aggregated emission per capita to small island destinations." is obvious.

The solution for the problem, namely, using Biofuel in place of traditional aviation fuel is substantiated in a limited way based on the assumptions made and the models the authors use. What is not discussed in the paper is the appropriateness of the choices the authors make. The bio fuel price is assumed to be 2-7 times that of conventional fuel, this  means very different scenarios to consider and not that the extra cost of implementing biofuel will vary between the amounts shown in Table 4,  last column. Mitigation solution can not be justified based on these figures alone. There are other costs and implementation issues and constraints which are not considered. 

The paper fails to connect the suggested solution and the case study emphatically.

Author Response

Response to Reviewer 2 Comments

Point 1: The main conclusion from the study, namely, "Distance is the determining factor of the emission per capita caused by international air transport, while the component of tourist origin can influence the aggregated emission per capita to small island destinations." is obvious.

Response 1: Thank you. We have revised this conclusion with more specific number. In the conclusion, Line 356, it has changed to " Distance is the determining factor of the emission per capita caused by international air transport, while the component of tourist origin decreases the aggregated emission per capita to small island destinations by 0.5%-2%.".

Point 2: The solution for the problem, namely, using Biofuel in place of traditional aviation fuel is substantiated in a limited way based on the assumptions made and the models the authors use.

Response 2: Thank you. We are aware of the limitation in biofuel production. However, due to current knowledge and the stage of the biofuel technology, it is not possible to come out with a very exact and quantitative result. In this study, assuming biofuel have the ability to reduce around 80% CO2 emission compared to conventional jet fuel, we estimated the biofuel demand for emission mitigation and obtained results of its corresponding cost, which is totally plausible. Besides, considering the limitation of our study, we have added uncertainty in the discussion from the aspects of fuel consumption, scenario assumption and biofuel price.

At the begin of 4. Discussion, we added " In fact, there are four main mitigation pathways for air transport emissionsuch as mitigation policy, technology improvement, operation and management, and alternate jet-fuel, within which, the contribution of operation and management is very limited. Besides, considering the commercial cost of the aircrafts, it is not a realistic solution to upgrade the aircraft very quickly unless the improved technology in airplane design or engine regeneration. As a result, other mitigation methods besides biofuel will not create significant changes in the coming century [22]. The European Aviation Environmental Report 2019 also states that sustainable aviation fuels have the potential to make an important contribution to mitigating the current and expected future environmental impacts of aviation [23]. For this reason, biofuel was considered as the major factor responsible for future emission mitigation. However, till now the potential of biofuel and the detailed approach, are yet to be discussed thoroughly. This study also suffers from a lot of uncertainties, arising from the prediction of emission and corresponding cost."

Point 3: What is not discussed in the paper is the appropriateness of the choices the authors make. The bio fuel price is assumed to be 2-7 times that of conventional fuel, this means very different scenarios to consider and not that the extra cost of implementing biofuel will vary between the amounts shown in Table 4, last column. Mitigation solution can not be justified based on these figures alone. There are other costs and implementation issues and constraints which are not considered.

Response 3: Thank you. As we reviewed in the introduction, the study on potential of biofuel for replacing the air transport emission is very limited. It is not possible to come out with a very exact and quantitative result. The reason we choose the assumption of biofuel price is 2-7 times of conventional fuel is that this is value provided by IATA sustainable aviation fuel roadmap (2015), and widely the accepted by general. Therefore, we have revised the discussion adding the uncertainty of the bio fuel price in Line 341 "Here, it was assumed that the biofuel price is 2-7 times of conventional fuel, but there are other ratio assumptions about the biofuel price. For instance, the European Aviation Environmental Report 2019 assumed that the biofuel price is 1.6~1.7 times that of conventional fuel. Furthermore, Pavlenko N stated that the price of alternative jet fuels is two to eight times the price of petroleum jet fuel. in the European Union [26]. Therefore, the cost estimated in this study contains a lot uncertainty, which can be reduced by further narrowing the range of the bio fuel price."

Point 4: The paper fails to connect the suggested solution and the case study emphatically.

Response 4: Thank you. But we respectfully disagree with your point. The case study of the two islands is an attempt to explore the biofuel mitigation pathway and its corresponding cost by a bottom-up method. Since Palau and Seychelles are two tourism-reliant islands, the article analysed the international air transport emission based on the international tourist perspective, and set three scenarios to seek the solution to realize the mitigation targets. Therefore, we are confirmed that the suggested solution (biofuel implements) and case study (two tourism-reliant islands emission mitigation) are connected. What's more, we have polished the article and the certification is attached in the PDF file.

Round 2

Reviewer 1 Report

All comments have been taken into account. The article has gained scientific value and can be a contribution to science but also has interesting information for practitioners. The method used is common and not really new, but the article itself meets the common requirements of scientific articles

Author Response

Response to Reviewer 1 Comments

Point 1: All comments have been taken into account. The article has gained scientific value and can be a contribution to science but also has interesting information for practitioners. The method used is common and not really new, but the article itself meets the common requirements of scientific articles

Response 1: Thank you. We are glad the last version responded to your comments properly. Thank your again for your contribution to the improvement of this article.

Reviewer 2 Report

Thank you for revising the paper based on the referee's report. Estimating the cost of biofuel mitigation path way is interesting. Biofuel mitigation path way does not consider implementation issues relating to the case, so the suggested solution is one among the alternatives available to realize mitigation targets. But this alternative is feasible, best and implementable, with respect to the islands discussed in the case study is not brought out. That is why I consider it is not emphatically justifying the claim made in the title. 

A suitable title could be, something like: 
Estimating the Cost of Biofuel Use to control Air Transport Emissions: A Case Study in Palau and Seychelles

Author Response

Response to Reviewer 2 Comments

Point 1: Thank you for revising the paper based on the referee's report. Estimating the cost of biofuel mitigation path way is interesting. Biofuel mitigation path way does not consider implementation issues relating to the case, so the suggested solution is one among the alternatives available to realize mitigation targets. But this alternative is feasible, best and implementable, with respect to the islands discussed in the case study is not brought out. That is why I consider it is not emphatically justifying the claim made in the title. A suitable title could be, something like: Estimating the Cost of Biofuel Use to control Air Transport Emissions: A Case Study in Palau and Seychelles

Response 1: Thank you for your suggestion. We are glad the purpose of the article is clear after we made the revisions based on your comments. The title you recommend is suitable for this article. Therefore, we changed our title into “Estimating the Cost of Biofuel Use to Mitigate International Air Transport Emissions: A Case Study in Palau and Seychelles”. Thank your again for your contribution to the improvement of this article.
